# Validation of the Spanish Versions of FACIT-PAL and FACIT-PAL-14 in Palliative Patients

**DOI:** 10.3390/ijerph191710731

**Published:** 2022-08-29

**Authors:** Estefanía Moldón-Ballesteros, Inés Llamas-Ramos, Jose Ignacio Calvo-Arenillas, Olaia Cusi-Idigoras, Rocío Llamas-Ramos

**Affiliations:** 1Nursing School of Zamora, University of Salamanca, Av. de Requejo, 33, 49022 Zamora, Spain; 2Virgen de la Concha Hospital of Zamora, Av. de Requejo, 35, 49022 Zamora, Spain; 3Nursing and Physiotherapy Faculty, University of Salamanca, Avda./Donantes de Sangre s/n, 37007 Salamanca, Spain; 4University Hospital of Salamanca, Paseo de San Vicente, n 182, 37007 Salamanca, Spain; 5Institute of Biomedical Research of Salamanca (IBSAL), 37007 Salamanca, Spain; 6Social Psychology Department, University of País Vasco, Barrio Sarriena s/n, 48940 Lejona, Vizcaya, Spain

**Keywords:** palliative care, quality of life, cancer, FACIT scales, validation

## Abstract

Palliative patients require several types of care to improve their quality of life as much as possible, and valid and reliable assessment instruments are essential. The objective of this study is the Spanish validation of the Functional Assessment Chronic Illness Therapy-Palliative Care (FACIT-PAL) and its abbreviated version, FACIT-PAL-14, in palliative care patients. FACIT-PAL and FACIT-PAL-14 were translated into Spanish and administered to 131 terminal oncology patients in home palliative care units, hospital palliative care units, health center teams, and social health centers. The European Organization for Research and Treatment of Cancer questionnaire, EORTC-QLQ-C15-PAL version, was used to evaluate the criterion validity. The EORTC-QLQ-C15-PAL was employed as a “gold standard”, and it obtained significant results with FACIT scales. FACIT-PAL-14, FACIT-PAL, and its subscales reported high internal consistency, from 0.640 to 0.816. The exploratory factor analysis for FACIT-PAL-14 found a structure in three factors that explained the 70.10% variance, and the FACIT-PAL scale found a structure of five factors. Physical wellbeing from FACIT-PAL highly correlated to the EORTC-QLQ-C15-PAL (r = 0.700), but social and family wellbeing was correlated to a lesser extent (r = −0.323). FACIT-PAL and the TOI (Toi Outcome Index) were also highly correlated with the EORTC-QLQ-C15-PAL, with values of r = −0.708 and r = −0.709, respectively. The Spanish versions of FACIT-PAL and FACIT-PAL-14 were demonstrated to be valid and reliable scales in palliative care patients.

## 1. Introduction

The industrial and technological revolutions, as well as the great increase in scientific discoveries, that have occurred in recent years have led to a worldwide epidemiological change. This has been supposed to an enlarge life expectancy, with its consequent chronification and more incidence of certain diseases. In this context, the need for adequate resources in palliative care (PC) is increasing [1,2].

The World Health Organization (WHO) places oncological diseases among the five most prevalent chronic diseases in the world. Cancer is postulated to be the main barrier preventing life expectancy increase, and it will increase during the 21st century [3,4].

An important aspect for these patients is quality of life, which helps us to know the global state of a patient, as well as the quality of the services offered. In the literature, there are reviews that identify the instruments developed and validated for quality of life; however, it is not possible to identify a study that assesses the processes of development, translation, cultural adaptation, and validation of such instruments in palliative care [5].

The use of standard and validated protocols is very useful for identifying a patient’s needs. In the scientific community, there are different instruments to assess the quality of life of cancer palliative patients [6].

The Functional Assessment Chronic Illness Therapy-Palliative Care (FACIT-PAL) and its abbreviated version, FACIT-PAL-14, evaluate the quality of life in palliative patients [7,8]. To have these instruments in Spanish can help clinicians in palliative care units. These scales have been translated in other languages, such as Turkish [9] and African [10].

Having the right instruments to assess our patients is essential. The high prevalence of cancer in our country means that special attention is paid to oncology patients. The FACIT-PAL scale and its short version, FACIT-PAL-14, have been validated in their original formats, demonstrating good validity and reliability. Therefore, their adaptation and validation into Spanish is proposed to have tools that allow us to know the quality of life of our patients.

The main objective is to validate Spanish versions of the FACIT-PAL and its abbreviated form, FACIT-PAL-14, from the original versions in English.

## 2. Materials and Methods

### 2.1. Participants

The study was performed from April 2018 to February 2021. The lack of data to calculate the sample size forced us to consider the sample sizes established in previous studies. Participants were selected following the next criteria.

Inclusion criteria

Patients older than 18 years who attended a hospital palliative care unit, health center team, social health center, or home palliative care units who presented a diagnosis of cancer for which they were receiving palliative care and who agreed to sign the informed consent before beginning the study were included.

Exclusion criteria

Patients were excluded who presented cognitive impairment or any neuropsychological disability that prevented them from answering the questionnaires, who had severe hearing loss or blindness, who did not know how to read or write, and who did not complete more than 75% of the scales.

### 2.2. Procedures

Permission for the use of the FACIT-PAL and FACIT-PAL-14 scales was requested through the official website [11]. The translation of the instruments from the original in English into Spanish was performed by a sworn translator of English and Spanish from the University of Salamanca. Subsequently, the opinion of a group of experts belonging to the Palliative Care Service of the Zamora Assistance Complex was collected to verify the suitability of the scales and to adapt certain terms. They suggested changes regarding the cultural meanings that were accepted. After that, we performed the back-translation method to translate it back into English and compare it with the original one, and no conceptual differences were found. Finally, the scale was applied to twenty patients to establish a definitive version as a pilot study to obtain the final version used in the validation study.

### 2.3. Data Collection

Data were collected using the following instruments:▪The EORTC-QLQ-C15-PAL questionnaire was developed by the European Organization for Research and Treatment of Cancer (EORTC) with the purpose of measuring the quality of life of patients in palliative care [12,13]. The scale was developed from the EORTC-QLQ-C30 questionnaire, which was created with the intention of measuring the quality of life of cancer patients [14,15]. The EORTC-QLQ-C15-PAL consists of 15 items. The first 14 collect information on different aspects related to the most frequent symptoms in palliative patients, and a global health scale corresponds with item 15.▪The Functional Assessment of Chronic Illness Therapy (FACIT) refers to an established and comprehensive set of tools aimed at measuring the quality of life of patients with chronic diseases [16]. This measurement system began with the creation of a generic questionnaire called “The Functional Assessment of Cancer Therapy—General (FACT-G)”. Based on this questionnaire, many specific FACIT tools have been developed for different pathologies, including the FACIT-PAL, a specific tool for palliative care, and its abbreviated version of 14 items, the FACIT-PAL-14 [7,8]. The FACIT-PAL scale consists of 46 items (Appendix A) that are divided into five subscales: physical wellbeing, social and family wellbeing, emotional wellbeing, and functional wellbeing, and the last one, which collects other additional concerns. The scale makes it possible to obtain a quality of life score according to each subscale or through three global scores: the FACIT-PAL Trial Outcome Index, or TOI, summarizes the index of physical and functional results; the Functional Assessment of Cancer Therapy—General, or FACIT-G, is obtained by adding the physical wellbeing index, the social and family wellbeing index, the emotional wellbeing index, and the functional wellbeing index; and FACIT-PAL total score provides more specific information on the wellbeing of the subject in the field of palliative care and is obtained by adding the 5 subscale indices.

All instruments were applied at the same time and were completed once.

### 2.4. Data Analysis

Microsoft Excel programs (version 2013, Redmond, WA, USA) and the IBM-SPSS statistical program (version 26, Chicago, IL, USA) were used.

A descriptive analysis of the sample was carried out. To know the psychometric properties of the scales, Cronbach’s alpha coefficient was used to assess reliability, and an exploratory factor analysis and Pearson correlations were used to assess validity [17,18].

### 2.5. Ethical Considerations

All participants were verbally informed of the study methodology and its objectives and were given an information sheet, as well as an informed consent, which they signed to participate.

All records were made only in writing, and the questionnaires lacked personal data.

All data were guarded and processed according to Organic Law 3/2018 of December 5 on the Protection of Personal Data. Likewise, the information obtained in this study was used exclusively to carry it out.

This project was approved by the Clinical Research Ethics Committee of the Zamora Health Area on 27 March 2018, with reference number 415/P.I. TESIS 2018.

## 3. Results

### 3.1. Sample

The sample contacted was 275 patients. The inclusion criteria were reached by 262 patients, but 61 presented one of the exclusion criteria (46 had a cognitive impairment, 3 could not read or write, 8 suffered from hearing loss, and 4 were blind), and 70 refused to participate. The final sample was 131 patients (Figure 1).

A total of 52.7% (*n* = 69) of the sample was female, and the average age was 78.88 (SD 11.711) years; 40.5% (*n* = 53) were married, and 92.2% (*n* = 130) were retired (Table 1).

Most of the participants completed the questionnaires themselves (*n* = 123), while others did them with the help of a researcher or a third party (*n* = 8). The response rate of the scales was 100%, except for items Q1 (“*Regardless of your current level of sexual activity, please answer the following question. If you prefer not to answer it, please mark this box and go to the next section*”) and GS7 (“*I am satisfied with my sex life*”) related to sexual life, which were only answered by 15.72% of the sample.

In the recruitment phase, items GS6 (“*I feel close to my partner (or the person who is my main support*”)) and Pal12 (“*I am able to make decisions*”) required explanations for their understanding, and item GS7 (“*I am satisfied with my sex life*”) was not answered by most of the sample.

Participants reported that the scales included the most relevant symptoms of their processes. However, two participants reported that they would modify item Pal5 (“*I am constipated*”) to an item that includes all intestinal symptoms, as well as diarrheal processes, and that they would add items relate.

### 3.2. Reliability

The data reported high internal consistency for FACIT-PAL-14 and the FACIT-PAL subscales, with values ranging from 0.640 to 0.816 (Table 2).

### 3.3. Validity

▪Construct validity. An exploratory factor analysis found a structure in three factors that explained the 70.10% variance (Table 3).

▪Criterion validity: The EORTC-QLQ-C15-PAL was employed as a “gold standard”, and Pearson correlations were performed. Correlations between the FACIT-PAL and EORTC-QLQ-C15-PAL ranged from r = −0.323 for social and family wellbeing to r = −0.709 for the TOI. Physical wellbeing from the FACIT-PAL was also highly correlated with the EORTC-QLQ-C15-PAL (r = 0.700). These correlations are summarized in Table 4.

## 4. Discussion

Valid and reliable instruments are essential to evaluate and establish the treatment of palliative patients. The validation into the Spanish language of the FACIT-PAL and the FACIT-PAL-14 scales was the aim of this study.

The final sample was 131 participants. Other studies have had different samples sizes, as in the Colombian validation with 10 participants [19], the study performed in England with 256 participants [20], the Turkish study [9] recruiting 232 participants, another study developed in three African countries [10] recruiting 461 participants, and a Canadian study [21] with 116 participants.

Considering the “Castile and Leon Palliative Care Plan 2017–2020” [22] in the province of Zamora, where the present study was developed, 372 patients received palliative care due to an oncological disease. For that reason and taking into account the difficulty of sample size calculation, the sample size obtained could be considered representative of the population.

The mean age of the sample was considerably higher (78.88 years; SD 11.711). This may be because the province of Zamora is a territory with a very aged population, where the population pyramid is completely inverted. In addition, it was observed that the older population was more willing to participate, while young people were more overprotected by their relatives and health workers and considered that participating in the study could harm them. Agreeing with this, most of the participants were widowed and retired. In addition, the tumor with the highest incidence was colorectal, with 33.8% of the cases. The province of Zamora is among the provinces with the highest incidence of colorectal cancer in Spain, with an incidence of 65 cases per 100,000 inhabitants in 2018 [23].

### 4.1. Reliability

In the study carried out by Shinall et al. in 2018 [24] for the validation of the FACIT-PAL-14 scale, the psychometric properties were evaluated, and the Cronbach’s alpha index obtained was 0.76; in the present study on the Spanish version, the value obtained was 0.807. In this study, the FACIT-PAL subscales obtained values ranging from 0.640 to 0.816. These values are similar to other validation studies, such as the original version [20] with values ranging from 0.75 to 0.94, the Turkish version [9] from 0.732 to 0.94, and the African version [10] from 0.78 to 0.90 (Table 5).

### 4.2. Validity

In the Spanish validation study, the EORTC-QLQ-C15-PAL was used to establish correlations and the Kaiser–Meyer–Olkin value was 0.762. In addition, no studies were found to evaluate the construct validity of this scale similar to our study (Table 6). In the English study [20], the Kaiser–Meyer–Olkin value was 0.86. For criterion validity, the subscales were correlated with the ESAS (Edmonton Symptom Assessment System) and the CES-D (Center for Epidemiologic Studies Depression Scale). In both studies, the TOI, FACT-G, and FACIT-PAL subscales obtained similar values, although they were slightly higher in the Spanish version. The subscale of social and family wellbeing was the one that obtained the least value in both studies. For the subscale of additional concerns, the value obtained in our work was lower than that in the English one [20].

In the Turkish study, the symptoms of the ESAS scale and the KPS scale (Karnofsky performance status) were used as the “gold standard”. The results obtained also showed a good correlation of the scale. The Turkish study [9] also calculated an exploratory factor analysis, and the Kaiser–Meyer–Olkin value was 0.892.

### 4.3. Quality of Life

The evaluation of quality of life remains an important issue in this population, and personalized treatments must be implemented [25]. Cancer patients such as those in our study, have a perception of their quality of life that is influenced by the repercussions of the health process [26]; for that reason, the need for valid and reliable scales to measure their symptoms and to improve their wellbeing is essential. Additionally, they have a positive effect on their caregivers because, as Krug et al. exposed, there is a positive correlation between a patient’s quality of life and the burden of their caregivers [27].

In the Spanish study, the mean quality of life reported on the FACIT-PAL-14 scale was 24.91, slightly lower than the results found in another study [24] in which the mean quality of life was 32.5 in men and 31.7 in women. Regarding tumor type, the tumors with the highest quality of life scores were hematological (30.25) and endocrine (30). Gynecological (18.6) and gastric (21.7) tumors obtained the lowest scores.

For the FACIT-PAL scale, the means obtained in the subscales were 59.35 in the TOI; 46.69 in the FACT-G, and 84.27 in the FACIT-PAL. In the English study [20], the mean value obtained in the TOI was 91.8; that of the FACT-G was 75.3; and that of the FACIT-PAL was 132. In the Spanish version, the values of the subscales obtained were lower. This may be because the mean of our population was higher than those of the others, so the quality of life of the baseline patients may be lower.

### 4.4. Limitations

The greatest difficulties found when carrying out the study were with the recruitment of the sample due to the delicacy of this type of patient. When we were faced with a small sample, in some cases the number of participants for each type of tumor was not very large. In addition, patients with hearing loss or blindness, as well as those not knowing how to read, were excluded, and this exclusion could cause bias in the results because an important group of our population was not included in the study.

## 5. Conclusions

The Spanish versions of the FACIT-PAL and FACIT-PAL-14 were demonstrated to be as valid and reliable in oncology palliative patients as others previously validated and as the EORTC-QLQ-C15-PAL. They are useful to evaluate quality of life in palliative patients.

## Figures and Tables

**Figure 1 ijerph-19-10731-f001:**
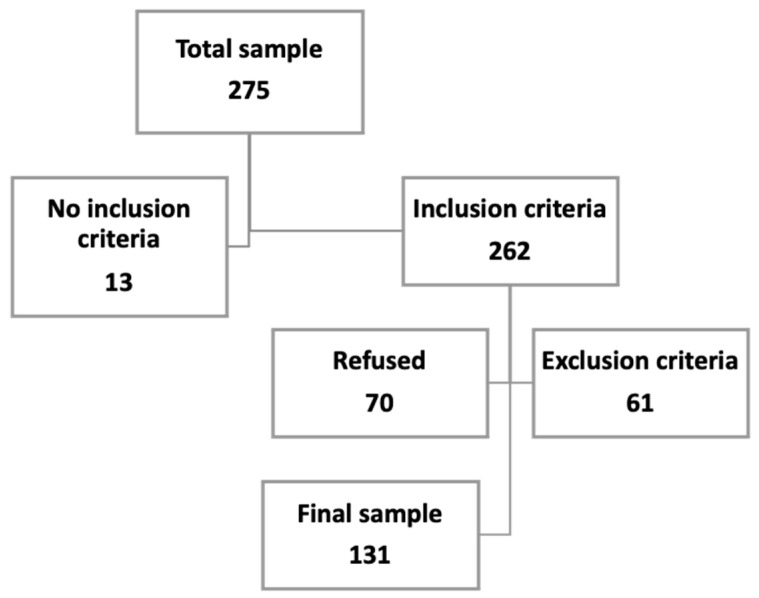
Sample size.

**Table 1 ijerph-19-10731-t001:** Characteristics of the sample.

Variable	*n* (%)	Mean (SD)
Age, years		78.88 (11.711)
Sex		
Male	62 (47.3%)	
Female	69 (52.7%)	
Place of recruitment		
Social health center	38 (27.5%)	
Primary care	43 (32.8%)	
Hospital	30 (22.9%)	
Home palliative care unit	22 (16.8%)	
Marital Status		
Single	22 (16.8%)	
Married	53 (40.5%)	
Divorced	7 (5.3%)	
Widowed	49 (37.4%)	
Level of education		
Primary school	99 (75.6%)	
Secondary school	3 (2.3%)	
High school	3 (2.3%)	
University	26 (19.8%)	
Occupation		
Worker	0	
Temporary disability	1 (1.4%)	
Housewife	0	
Unemployed	0	
Retired	130 (99.2%)	
Place of residence		
Rural	83 (63.4%)	
Urban	48 (36.6%)	
Cancer type		
Cerebral	5 (3.8%)	
Colon-Rectum	32 (24.4%)	
Endocrine	1 (0.8%)	
Stomach	10 (7.6%)	
Kidney	9 (5.6%)	
Larynx	5 (3.8%)	
Leukemia	2 (1.5%)	
Breast	11 (8.4%)	
Melanoma	1 (0.8%)	
Myeloma	2 (1.5%)	
Neurologic	5 (3.8%)	
Pancreas	7 (5.3%)	
Prostate	5 (3.8%)	
Lung	19 (14.5%)	
Sarcoma	2 (1.5%)	
Thyroid	1 (0.8%)	
Uterus	5 (3.8%)	
Vaginal	2 (1.5%)	
Bladder	3 (2.3%)	
Time since diagnosis		
1 year	39 (29.8%)	
2 years	46 (35.1%)	
2–3 years	1 (0.8%)	
3–5 years	33 (25.2%)	
5–10 years	11 (8.4%)	
>10 years	1 (0.8%)	
Main caregiver		
Son or daughter	10 (7.6%)	
Son or daughter with partner	29 (22.13%)	
Son or daughter with other	41 (31.3%)	
Other	28 (21.4%)	
Partner	2 (1.5%)	
Partner, son or daughter, and other	19 (14.5%)	
Partner with other	2 (1.5%)	
Time to complete scales		
FACIT-PAL-14		1:47 (00:12) min
FACIT-PAL		4:21 (00:23) min

FACIT-PAL: Functional Assessment Chronic Illness Therapy-Palliative Care.

**Table 2 ijerph-19-10731-t002:** Internal consistency with Cronbach’s alpha coefficient.

Scale/Subscale	Cronbach’s Alpha
FACIT-PAL-14	0.807
FACIT-PAL	0.751
Physical wellbeing subscale	0.816
Family wellbeing subscale	0.640
Emotional wellbeing subscale	0.784
Functional wellbeing subscale	0.816
Additional concerns subscale	0.712

FACIT-PAL: Functional Assessment Chronic Illness Therapy-Palliative Care.

**Table 3 ijerph-19-10731-t003:** Exploratory factor analysis for FACIT-PAL-14.

Rotated Component Matrix ^a^			
	Component		
	1	2	3
GE1 (I feel sad)	0.737	−0.339	
GP2 (I have nausea)	0.718		−0.303
GP4 (I have pain)	0.701		
Pal5 (I am constipated)	0.639		
GE6 (I worry that my condition will get worse)	0.605		0.356
Pal4 (I feel like a burden to my family)	0.532		
B1 (I have been short of breath)	0.519		
GP1 (I have a lack of energy)	0.500	−0.406	
GF5 (I am sleeping well)	−0.500		
GF3 (I am able to enjoy life)		0.763	
GF7(I am content with the quality of my life right now)		0.719	
Sp21 (I feel hopeful)		0.701	
GS2 (I get emotional support from my family)			0.838
Pal14 (I am able to openly discuss my concerns with the people closest to me)		0.476	0.615

Extraction method: principal component factor analysis. Rotation method: standardized varimax with Kaiser. ^a^. The rotation converged in 5 iterations.

**Table 4 ijerph-19-10731-t004:** Criterion validity with EORTC-QLQ-C15-PAL.

Scale	Pearson Correlations
FACIT-PAL-14	0.496
FACIT-PAL	
Physical wellbeing subscale	0.700
Family wellbeing subscale	−0.323
Emotional wellbeing subscale	0.601
Functional wellbeing subscale	−0.531
Additional concerns subscale	0.171
TOI	−0.709
FACT-G	−0.699
FACIT-PAL INDEX	−0.708
GENERAL	0.497

FACIT-PAL: Functional Assessment Chronic Illness Therapy-Palliative Care. TOI = Trial Outcome Index.

**Table 5 ijerph-19-10731-t005:** Reliability comparison.

Study	Cronbach’s Alpha Coefficient General	Cronbach’s Alpha Coefficient Physical Wellbeing	Cronbach’s Alpha Coefficient Family Wellbeing	Cronbach’s Alpha Coefficient Emotional Wellbeing	Cronbach’s Alpha Coefficient Functional Wellbeing	Cronbach’s Alpha Coefficient Additional Concerns
English [21]	0.94	0.85	0.75	0.80	0.84	0.82
Turkish [9]	0.932	0.732–0.860				
African [10]	0.90	0.83	0.78	0.80	0.87	0.81
Spanish	0.751	0.816	0.640	0.784	0.816	0.712

**Table 6 ijerph-19-10731-t006:** Validity of FACIT-PAL scale.

Study	Kaser–Meyer–Olkin (KMO) Value	Bartlett Index	Chi-Squared Test Value	Exploratory Factor Analysis Structure
Spanish	0.762	<0.05	2551.32	5 Factors
English [21]	0.86	<0.05	2773.66	5 Factors
Turkish [9]	0.892	<0.05	5103.4	5 Factors

## Data Availability

All data are available upon reasonable request to the corresponding author.

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
