# Peer review of "Validation of the Spanish Versions of FACIT-PAL and FACIT-PAL-14 in Palliative Patients"

_ijerph, 2022, doi:10.3390/ijerph191710731_

Round 1

Reviewer 1 Report

The research paper entitled "Spanish Versions of FACIT-PAL and FACIT-PAL-14 in Palliative patients (Manuscript ID: ijerph-1834553)” was reviewed. After reading the manuscript, I suggest to author to revise wisely for publication in International Journal of Environmental Research and Public Health. Please do the revision for this manuscript based on comments below:

1. The abstract should be re-written to summarize the work; the abstract should state briefly the purpose of the research, the PRINCIPLE results and MAJOR conclusions. An abstract is often presented separately from the article, so it must be able to stand alone.

2. The novelty of this study compared to other studies is not clear.

3. Authors must compare their results (in a Table) with others reported in the literature. The mentioned tables are not complete.

4. The comparison with the results of others in the past is very low.

5. The authors must revise the manuscript carefully to eliminate grammatical errors and typo-errors.

Author Response

Thank you very much for your time to review our work and for your suggestions. We answer them point by point.

  1. Abstract has been rewritten.
  2. The novelty of this study is the scale validation, these scales couldn’t be used in our language due to the lack of this process until now. Unfortunately, there are few previous studies published to compare.
  3. In table 5, we have showed the reliability of the other studies which have been validated these scales to establish comparisons. Do you need any more comparisons?
  4. Sorry about this concern, unfortunately, discussion has been written regarding the few articles published about this topic.
  5. All manuscript has been reviewed again to correct grammatical errors.

Thank you for helping us to improve our manuscript.

Reviewer 2 Report

As the manuscript is just a scale validation in Spanish, there is a lack of conceptual and theoretical contribution to the field. The style of writing is like a student report without much new ideas not points of interest.

Author Response

Thank you for your comment and time dedicated to this review. We are sorry about this opinion, the need of validated tools to treat our patients is essential, the validity of this scale, allow us to measure their symptoms to improve their quality of life. We hope you like the changes we have made.

Reviewer 3 Report

Intro - need to rewrite the aim 

Procedure - the translation procedure is not complete < did you do back translation, how many translated etc

Need to clarify in the method about pilot portion and the results 

line 115-116 need to clarify the itmes 

need to justify sample size 

Table 1 n=25 ???

need to regroup cancer types 

3.2 line 154-164 need to clarify the paragraph its out of line 

table 3 abbreviations are not explained 

Discussion 

line 190-194 need to decide about sample size and justification 

all section need to be rewritten 

Author Response

Dear reviewer, thank you for your time dedicated to improve the quality of our manuscript. We answer your suggestions point by point.

  1. Intro - need to rewrite the aim 

Thank you, the aim has been rewritten.

  1. Procedure - the translation procedure is not complete < did you do back translation, how many translated etc

Thank you very much for your suggestion. This section has been clarified.

  1. Need to clarify in the method about pilot portion and the results 

Thank you for your consideration, this explanation has been added in the procedure section.

  1. line 115-116 need to clarify the itmes 

Thank you for your considerations, items are explained in the supplementary material.

  1. need to justify sample size 

Thank you for your comment, this information has been added in the results section.

  1. Table 1 n=25 ???

Sorry about that, it was a mistake

  1. need to regroup cancer types 

Thank you for your suggestion, unfortunately at this point we disagree, we know that there are few cases of some types of cancer which prevent us to establish comparisons but we think that all of them are important. Also, we prefer showing all types of cancer instead regroups because some types cancers loss their visibility.

  1. 2 line 154-164 need to clarify the paragraph it’s out of line 

Thank you very much for your consideration. This paragraph has been deleted.

  1. table 3 abbreviations are not explained 

Thank you. You are right, but it is the code of the items of this scale, the abbreviation explanation has been added in the table.

  1. Discussion 
    • line 190-194 need to decide about sample size and justification 

Thank you for your suggestion. This information has been added.

  • all section need to be rewritten 

Thank you very much for your comments. These sections have been rewritten

Reviewer 4 Report

There is an error in the abstract. Where it says FACIL-PAL (line 19) it should put FACIT-PAL

The main objective does not describe the depth of what is intended to be carried out. If the objective is to make a translation only, then the psychometric properties of the scales would not have been the main body of the study, and the conclusions do not refer to the translation (logically, because the translation should not be the objective of the investigation, but a necessary step)

Secondly. Considering this same section. If what is intended is to make a translation, the method carried out is not the correct one. Today, what is expected is a translation plus a back translation plus a new translation, all carried out by experts in the target language and also in the field of study (steps can be seen in Beaton et al. 2000 - Guidelines for the Process of Cross-Cultural Adaptation of Self-Report Measures). These steps have not been carried out

In lines 82 to 85 it is said that the opinion of experts was taken into account and the questionnaire was also administered to a sample of patients. In both cases, both the contributions made by both groups (problems found and solutions) should be described, as well as the rubric used so that the reader knows exactly what instructions these reviewers had and what they had to pay attention to specifically.

As for the results:

1º I do not understand why in table 4 the correlations appear with a negative sign and in the explanation they are spoken of as positive correlations.

2nd Point 3.2. Quality of life. Lines 160 to 164. This paragraph is not in context. It is not understood what it means. If you intend to make a comment about the results, it should go in the discussion section.

3º You show a factorial analysis that shows three factors, which do not correspond to the original scales. On the other hand, in the discussion you refer to the KMO of other studies, but you have not calculated it. Would not a confirmatory factor analysis have been more appropriate than an exploratory one that, moreover, does not replicate the theoretical model?

Finally, we cannot conclude that the questionnaire is valid and reliable. Although it is reliable, the validation carried out is partial, so we could talk about partial validation and reliability.

Author Response

Dear reviewer, thank you for your time dedicated to improve the quality of our manuscript. We answer your suggestions point by point.

  1. There is an error in the abstract. Where it says FACIL-PAL (line 19) it should put FACIT-PAL

You are right, thank you, the mistake has been corrected.

  1. The main objective does not describe the depth of what is intended to be carried out. If the objective is to make a translation only, then the psychometric properties of the scales would not have been the main body of the study, and the conclusions do not refer to the translation (logically, because the translation should not be the objective of the investigation, but a necessary step)

Thank you for your consideration, the objective has been rewritten.

  1. Considering this same section. If what is intended is to make a translation, the method carried out is not the correct one. Today, what is expected is a translation plus a back translation plus a new translation, all carried out by experts in the target language and also in the field of study (steps can be seen in Beaton et al. 2000 - Guidelines for the Process of Cross-Cultural Adaptation of Self-Report Measures). These steps have not been carried out

Thank you for your consideration. Translation method has been explained to a better understanding.

  1. In lines 82 to 85 it is said that the opinion of experts was taken into account and the questionnaire was also administered to a sample of patients. In both cases, both the contributions made by both groups (problems found and solutions) should be described, as well as the rubric used so that the reader knows exactly what instructions these reviewers had and what they had to pay attention to specifically.

Thank you very much, the information requested has been added to this section.

  1. As for the results:

1º I do not understand why in table 4 the correlations appear with a negative sign and in the explanation they are spoken of as positive correlations.

Thank you for your carefully review of our manuscript, it was a mistake that has been corrected.

2nd Point 3.2. Quality of life. Lines 160 to 164. This paragraph is not in context. It is not understood what it means. If you intend to make a comment about the results, it should go in the discussion section.

Thank you for your comment. This paragraph has been deleted.

3º You show a factorial analysis that shows three factors, which do not correspond to the original scales. On the other hand, in the discussion you refer to the KMO of other studies, but you have not calculated it. Would not a confirmatory factor analysis have been more appropriate than an exploratory one that, moreover, does not replicate the theoretical model?

Thank you for your suggestion. The confirmatory factor analysis would be more appropriate and it would be considered in future studies. We decided our analysis to be able to compare with other studies published because no one obtained this data. KMO values have been added.

  1. Finally, we cannot conclude that the questionnaire is valid and reliable. Although it is reliable, the validation carried out is partial, so we could talk about partial validation and reliability.

Thank you for your considerations. We hope the changes made from your comments improve the understanding and quality of our study.

Round 2

Reviewer 2 Report

I do not think a validation of scale in a specific language is of sufficient theoretical and conceptual sophisticated that reach the publication of this journal.

Author Response

Dear Reviewer 2,

First of all, thank you for your effort in reviewing this paper. All of the considerations are important and the reviewers feedback is the key of the manuscripts’ improvement.

Patient assessment is very important in the clinical setting and even more when dealing with health issues. However, not all instruments meet the psychometric criteria to be validated, and this important process is the key to their use. Without such validation process, the results obtained could be not reliable or reproducible. As the reviewer comments, validation in a single language is less important than an universal validation, but, as the reviewer will be aware, the validation process is not a mere translation, it involves a cultural adaptation, which is why a scale validated in Spanish in Spain is not the same as a scale validated in Spanish in any Latin American country, since despite being the same language, certain terms are different.

Reviewer 4 Report

I agree with the changes made. Be careful with line 220 they have not translated the item Cronbach's Alpha

Author Response

Dear Reviewer 4,

Thank you again for your considerations and effors in reviewing our manuscript.

Sorry about that, this term will be translated properly.